# Digital Marketing of Commercial Complementary Foods in Australia: An Analysis of Brand Messaging

**DOI:** 10.3390/ijerph18157934

**Published:** 2021-07-27

**Authors:** Trish Dearlove, Andrea Begley, Jane Anne Scott, Gemma Devenish-Coleman

**Affiliations:** School of Population Health, Curtin University, Perth 6102, Australia; patricia.dearlove@postgrad.curtin.edu.au (T.D.); a.begley@curtin.edu.au (A.B.); jane.scott@curtin.edu.au (J.A.S.)

**Keywords:** digital advertising, baby foods, complementary feeding, infant feeding, food marketing, online, promotion

## Abstract

The digital marketing of commercial complementary foods (CCF) is an emerging area of concern in Australia. Although research into traditional methods has identified a range of problems, the marketing and messaging strategies employed within digital spaces have gone largely unscrutinized. This study sought to examine the methods used by CCF manufacturers to promote Australian baby foods and brands in a digital space. A multiple step approach was used to assess the CCF brands available in major Australian retailers, the social media platforms they used, and to thematically analyze the text and visual messages contained in posts published over a three-month period. Of the 15 brands identified, 12 had a digital presence, and all of these used Facebook. Four themes emerged from an analysis of 216 Facebook posts; (1) general product attributes, (2) socially desirable attributes (which included messaging related to taste (41%), self-feeding (29%) and fun (19%)), (3) concern-based attributes (including organic status (40%), age targets (39%) and additive-/allergen-free status (18%)) and (4) health-focused attributes (which included messaging related to healthy/nutritious ingredients (45%), and child development/growth (15%). Messages contained in Facebook posts were mostly positive brand/product aspects (Themes 1 and 2) or parental concern-based aspects (Theme 3 and 4). These themes match previous analyses of marketing content in traditional media and should be closely monitored due to the personalized nature of consumer social media interactions.

## 1. Introduction

The importance of early life nutrition is well recognised in the development of later health outcomes, yet in Australia there are few regulations or policies that address infant and young child feeding (IYCF) practices other than breastfeeding [1,2,3]. The World Health Organization (WHO) Resolution on Ending the Inappropriate Promotion of Foods for Infants and Young Children provides recommendations to support optimal IYCF practices [4] however these are poorly implemented in Australia [5]. The Marketing in Australia of Infant Formulas: Manufacturers and Importers Agreement (MAIF) is focused exclusively on regulating infant and follow on formulas, which in Australia are those suitable from birth to 12 months of age [6]. Developed in response to the WHO International Code of Marketing of Breastmilk Substitutes, the MAIF agreement is self-regulated by signatories who have been criticized for their refusal to include toddler milks—formula-like milk products for children 12 months and above—and complementary foods as part of this agreement [6,7]. Additionally, while the 2019 National Breastfeeding Strategy details some recommendations for improvements in IYCF practices, they are few and poorly supported by practical policy or legislation [8]. Critically, much of the development of policy and recommendations is focused on breastfeeding and breast milk substitutes, including toddler milks, which leaves complementary feeding unscrutinized.

Complementary foods (CF), which are those first foods given to babies and toddlers alongside continued breastfeeding or infant formula, are recommended to be timely, adequate, safe and appropriate [4]. The WHO emphasizes the importance of CF being home prepared using nutrient-rich, locally available foods and advises not to introduce CF before six months of age [4]. In Australia, early introduction of solid foods is an issue, with approx. 35% of babies receiving solid food at 4 months of age and 92% by 6 months of age [9]. Recent research has highlighted the problematic nature of commercial complementary foods (CCF) which disregard the WHO recommendations across each of the four domains;

Timely—CCF packaging that implies suitability for infants under 6 months [10];Adequate—high added sugar content and generally low nutrient density of CCF render them inappropriate for the needs of infants [11,12];Safe—ready-to-eat CCF products are not sterile once opened and pose a risk if leftovers are not stored correctly, additionally, squeeze pouch caps pose a choking hazard [10];Appropriate—CCF from squeeze pouches delivers food via a sucking motion which, in excess, may hinder oral skill development [10]. Additionally, the smooth-puree texture of most ready-to-eat CCF is at odds with texture progression recommendations to aim for family foods by 12 months of age [13].

In recent years there has been a dramatic increase in the number of CCF available for purchase in Australia [14], including those designed as first solids through ready-to-eat meals in various packaging formats, to snack foods for toddlers and young children, as well as a rise in the marketing of these products. Many snacks marketed to infants and toddlers are classified as discretionary foods—energy dense, nutrient poor foods not recommended as part of a healthy diet for infants and young children [3,15]. 

Traditional media sources of advertising CCF and breast milk substitutes such as parenting magazines have been the subject of past research, while newer forms such as digital marketing are a more recent area of study. Research has centred on breast milk substitute promotion as well as the direct targeting of children and adolescents by the food industry [16,17,18,19,20,21,22]. Social media is emerging as an area of much concern in marketing due to the personalized nature of platforms which offer targeted content direct to consumers [23]. This personalized content, where ads appear to individuals based on previous buying or viewing history, has been associated with increased brand awareness and ad credibility as well as a reduced resistance to the ad [24]. Australians are enthusiastic users of social media—60% of the total population are active users on Facebook, half use YouTube and 35% of Australians are on Instagram [25]. Digital marketing has been embraced by industry as a means to shape parental perceptions and the social norms for IYCF [20,26], positioning their brands as providers of sympathetic support [23]. This is achieved by the extensive consumer research that goes into understanding the desires and drivers of parenting attitudes, allowing corporations to profile consumers and market brands that are targeted to appeal to the aspirations of the parents [23]. Parents have long been recognised as gatekeepers of children’s nutrition, yet the assumption that parents are able to determine what constitutes healthy eating behaviors is flawed [27]. The huge volume of marketing and mass media content leads to information fatigue and the disregarding of evidence-based nutrition advice from health professionals, which is seen as merely another opinion, less valued than the views of peers and family [27,28,29]. It is necessary, therefore, to understand what is being said to parents through marketing in order to combat the norms CCF manufacturers are trying to influence and the impact on IYCF this may have [30].

This study sought to understand what methods are being used by manufacturers of non-perishable CCF available in Australia to promote baby food brands and products in a digital space. The objectives were: to identify the CCF brands available in major Australian retailers and the digital media advertising platforms they use; to identify what the stated marketing messages present on selected social media are; and to determine what implicit messages are being conveyed by text and imagery of these sites to promote CCF.

## 2. Materials and Methods

A multiple step approach was used to obtain a three-month snapshot of explicit and implicit marketing messages.

### 2.1. Brand Audit

A brand audit identified CCF brands available for purchase in leading Western Australian supermarkets and pharmacies by searching the ‘baby food’ category of retailer websites. These are the shelf stable, bagged, boxed and jar foods typically displayed in the baby aisle of supermarkets and pharmacies alongside breastmilk substitutes. Coles and Woolworths, the two major supermarket chains in Australia, represent more than 67% of the Australian baby food retail market [14], thus were deemed suitably representative of the range of brands available. A Google search of pharmacies was conducted to determine if there were significant brands available by ordering online from these retailers. It was found that a majority of pharmacies carry few, if any, CCF products. Chemist Warehouse was the only exception, therefore this retailer was also included. No product age range was used (e.g., 0–12 months, 2+ years) as the products observed either had only a minimum age rather than a range (e.g., suitable from 8 months) or offer no age recommendation at all. While food for young children may be considered as designed for those under three years of age, it should be noted that there is no restriction preventing their consumption by, or promotion to older children.

The number and type of unique non-perishable CCF products available at each outlet was recorded from the retailer websites. Products were classified into four categories and five sub-categories; ready-to-eat meals (subdivided by packaging format: jar, squeeze pouch, larger pouch), ready-to-prepare foods (for example porridge and rice cereal), ready-to-cook (for example dry pasta) and ready-to-eat snack (subdivided into teething rusks or older baby/toddler snack). Infant formula and toddler milks were not included in this study. The number of CCF products available from each retailer for each brand was calculated as a total and by product type, and the relative market share of each brand was determined. 

### 2.2. Websites and Social Media—Quantitative Analysis

Brands that did not have a website or social media account were retained in the brand audit reporting, but were ineligible for the remaining analyses. Where available, Australian websites were located by a Google search to imitate consumer behaviour. Descriptive data were collected from these websites, including links to brand social media accounts. Facebook was chosen as the social media platform to be investigated due to its use by all brands, as well as having the highest number of users in Australia [25]. The number of likes and followers of each brand’s Facebook page were recorded, along with the total number of posts published in the preceding three months (10 June–11 September 2020). This was used to calculate the mean monthly posting rate for each brand. Due to the transient nature of information in online spaces, all posts in this period were captured using screenshots for thematic analysis.

### 2.3. Facebook—Qualitative Analysis

The marketing messages presented via Facebook posts by CCF manufacturers were thematically coded using a deductive approach. The first author (TD) developed the codebook and coded all data. Comments and other user generated content were excluded from coding unless they were reposted by the page administrator, i.e., only content that was created by the brand was coded. Hashtags were also excluded. A set of initial codes were identified a priori, based on prior research of traditional media [31,32], and preliminary researcher observations. These codes were then piloted using the first month’s posts (capped at 10) for each brand. During this stage, the code descriptions were refined, and additional codes were identified and added. This set was then used to code all Facebook posts that had been captured for each brand. A multiple-pass process was followed for each post. Firstly, the text of each post was reviewed for relevant keywords and an initial set of codes assigned. Secondly, any accompanying images were reviewed for messages not inherent in the post text. A final text review confirmed the code selection and identified any additional messages not initially noted. Posts were assigned as many codes as were applicable. During this process any messaging that did not fit the code categories were noted. Scrutiny of these notes resulted in the development of six further code categories and all data were analysed a second time. The final list of codes and their descriptions are presented in Appendix A. 

A second coder (AB) independently assigned codes to a sub-sample to assess bias. Inter-coder reliability was calculated using simple percent agreement of a sample of 20 posts selected by random number generation. Across the 20 posts, 233 codes were assigned, of which 210 (90.1%) were in agreement. Discrepancies were assessed independently by a third reviewer (GDC) and discrepant codes discussed and then consensually coded. Codes were then assessed for frequency and grouped into themes. Additionally, each post was assigned to one of seven categories based on content type (product range, broad values, social/parenting support, incentives, informative, engagement/connection building, expert/celebrity endorsement), descriptions of which are presented in Appendix A. All data coding and analysis were performed in Microsoft Excel (Microsoft, Albuquerque, NM, USA). 

## 3. Results

Across the 15 brands identified in the brand audit, there were 210 unique CCF lines in Coles, with smaller product ranges found in Woolworths and Chemist Warehouse (Table 1). No category of product was represented by all brands, though most produced products in the ready-to-eat categories, both meals and snacks, which were the most numerous product types, comprising over 80% of products sold in Chemist Warehouse and more than 90% of products in Woolworths and Coles. Ready-to-eat meals packaged in jars were exclusive to the Heinz brand and not found in Chemist Warehouse, while ready-to-cook baby foods, exclusive to Bellamy’s Organic brand were only found in Chemist Warehouse. 

Brands differed in the range of products offered across the retailers, with no outlet carrying all 15 brands. Table 2 shows the presence of each brand in each retailer. Major brands across all three outlets included Rafferty’s Garden (22–25% product range share), Heinz/Farex (16–20%) and Bellamy’s Organic (7–23%). Only Organic (13–18%) and Bellies (9–16%) were prominent in supermarkets, while Bubs Organics had a strong presence in the pharmacy chain (17%).

Of the 15 brands identified, 12 were found to have a website presence. One website featured two of these brands (Farex, owned by Heinz), so a total of 11 websites were investigated in the remaining analyses. The brands that did not have a website were all supermarket brands (Macro, Smiling Tums, CUB). Facebook (*n* = 11, 100%) and Instagram (*n* = 9, 82%) were the most common social media accounts linked to the brand websites, with YouTube (*n* = 4, 36%), Twitter (*n* = 2, 18%) and WeChat (*n* = 1, 9%) the other social media platforms used (Table 3).

Facebook page followers and likes were very variable but did not appear related to posting frequency in this measurement period. A total of 216 posts were captured across 10 brand pages, with Heinz the only brand to not produce any social media content in the monitoring period. Product range was the most common post type for 7 of the 11 brands and comprised nearly half of all post types (Table 4). Brand image building posts were also frequently evident in the form of content about broad company values (14.8%) and posts that sought engagement or connection with parents (14.8%). Incentives (that is, giveaways, competitions or other prize giving opportunities) comprised 10.6% of all brand posts but were the single most common post type for Baby Mum Mum (66% of Baby Mum Mum posts). With the exception of one Father’s Day engagement piece, all posts by Nestle (*n* = 5) referred consumers to their website, while other brands used this technique on average one in every ten posts.

Three of the 11 brands are also manufacturers of formula and toddler milk products, with only Nestle not featuring milks, nor any branded product, in their Facebook posts. Both Bubs Organics and Bellamy’s Organic promoted toddler milks heavily in their posts and both were the only brands to utilize celebrity endorsement. The use of endorsement by celebrities or experts was found in less than 5% of total posts; this technique was utilized by only three brands. A quarter of Bellies posts contained the advice of their ‘natural feeding expert’ to promote their ready-to-eat snacks. 

As shown in Table 4, positive images of babies and children appeared in almost half of all posts (48.1%). Messages around taste/flavor, organic status of products, self-feeding and healthy/nutritious ingredients also featured heavily in posts, often in combination. About 40% of posts (*n* = 84) contained messaging about a specific age target, usually a “suitable from *x* months” for the product. The appearance of packaging with the “4+ months” occurred in nine of the 216 posts, with all ready-to-eat snack food products carrying a minimum age of 6+ months. Bubs Organics was the only brand to specifically target an older child age range with their junior nutrition drink recommended for ages 3–12 years, although images of school-aged children were used by several brands.

Four themes emerged from the marketing messages of the Facebook posts: general product attributes, socially desirable attributes, concern-based attributes and health-focused attributes. General product attributes incorporated messages that made the product sound appetizing, around texture—“new texture adventures”, and taste—“yummy new flavors” as well as promoting novel products (e.g., mini yoghurt drops). Socially desirable attributes were the social ideals that promote fun and positivity for the child—“make snack time fun”, or ease for the parent in the form of a solution to a perceived problem—“soothe those teething troubles”, or promotion of self -feeding which has a dual role in being convenient for parents as well as popular within the ‘baby-led weaning’ approach. The promotion of Australian made or owned via use of ‘local’ suppliers or ‘Aussie mum and dad’ created businesses was also considered a social ideal message, as was positive imagery of babies and children. Concern-based attributes included those designed to appeal to a particular kind of consumer need such as organic ingredients, or the absence of allergens or additives—“free from GMOs, chemicals and preservatives”, for parents who value these. Health-focused attributes were the messages that target parental desires to ensure the health of children, through supporting growth and development—“grow up strong” and “encourage… tongue lateralization”, assisting health—“support their immune system” or developing healthy habits—“foster a healthy relationship with food”, as well as the promotion of healthy ingredients, specific nutrients and the nutritious quality of products—“premium nutrition enriched with prebiotics and probiotics”. A detailed list of messages and the type of product it promotes can be found in Appendix A. Broadly, messaging may be considered as falling into one of two domains; to either promote some aspect of the product deemed as desirable or to capitalize on some aspect of parental concern.

## 4. Discussion

This study investigated the baby food brands available in major Australian retailers, the social media advertising platforms they use, and the messages conveyed via language and imagery to promote CCFs in Australia. Facebook is utilized by Australian CCF manufacturers to build positive brand perceptions through posts designed to be emotionally engaging and supportive as well as tactics such as endorsements by celebrities or experts, competitions and giveaways. This positive branding also encompasses promotion of specific brand values and products. The four themes that emerged from the content analysis of messages published in Facebook posts can be seen as encompassing two domains. First, desirable attributes like taste, texture and self-feeding. Second, parental concerns targeting health, development, allergens, additives and organic ingredients, which may be thought of as fear-based drivers. While marketing of a brand or product’s desirable attributes is an expected advertising practice, the fear-based aspect is of concern due to the potential to promote misleading or unrealistic expectations regarding infant feeding. 

The effect of CCF marketing on consumer attitudes and behaviour is an area of concern, particularly via digital platforms which allow personalized, targeted advertising direct to parents and caregivers [33,34]. A 2020 systematic review of qualitative research regarding parent perceptions of infant feeding showed “beliefs, values, and perceived norms were a central influence on complementary feeding practices” [35]. The impact of such messaging that targets parental concerns such as fear of choking have been detailed in recent qualitative studies. An investigation of Australian mothers understanding of infant feeding found that although some mothers expressed guilt about using CCF, specific characteristics such as being organic increased the acceptability of these products [28]. For some mothers, CCF were considered safer than home-made foods because the Mothers lacked confidence in their own ability to prepare foods to the correct texture and amount, with age recommendations on CCF packaging viewed as most likely to be texturally appropriate, and therefore safe from choking [28]. Marketing messages reviewed by this study demonstrate that manufacturers are building their marketing around these concerns; promoting the ‘benefit’ of organic foods and smooth purees, or snacks “designed for little hands” to encourage self-feeding. This tactical marketing has the potential to direct parents away from home prepared foods, undermining confidence of parents and promoting confusion regarding the optimal age for introduction of solids [28]. 

Health consciousness, fueled by targeted marketing of infant nutritional needs is noted by industry as a key driver of market growth [14]. Marketing to a health conscious audience is evident in health-focused theme of this study, and also plays to parental concerns. The awareness of health as a priority for consumers extends past the promotion of healthy ingredients like fruits and vegetables, to claims of broader benefits that products help “foster a healthy relationship with food”. Similar to the marketing of toddler milks in Australia and the US [33], the highlighting of nutritious content in CCF—such as vitamins and minerals, pre and probiotics, dietary fibre, omega 6—may result in a ‘health halo’ effect [33] that leads to parents attaching benefits to these products that may not exist. Furthermore, it has been suggested that this type of marketing may diminish the value of home meals and work to persuade parents that costly commercial foods are essential [33]. Recent research on the nutritional and textural composition of CCF available for sale in Australia demonstrates their unsuitability as a primary food source, being predominantly sweet flavored, low in iron, snack foods that would be classified as non-core, or discretionary items, and purees lacking appropriate textural variety [15,36]. The importance of texture in the development of skills is utilized in marketing by brands promoting the value of their snack products in self-feeding and of attributes such as “tongue lateralization”, which encourages the perception that these products are necessary for optimal child development. Normalization of manufactured snack foods within the diet, whether they are classified as discretionary foods or not, may encourage high intakes of energy-dense snack foods in later life [15].

In Australia, health and nutrition content claims are prohibited from use for infant formula products, both packaging and advertising, under FSANZ Standard 1.2.7 [19]. It is recommended that this be extended to include infant foods and growing up milks, or any food specifically targeted at children. Additionally, the restriction of health halo statements is advisable. Further, FSANZ Standard 2.9.2-7 specifies that “The label on a package of food for infants must not include a recommendation, whether express or implied, that the food is suitable for infants under the age of 4 months” [2]. No clear breach of this policy was found in the marketing material studied. However, marketing of products as suitable ‘first foods’ without specifying from age 6 months, as well as packaging indicating suitability from 4 months of age promotes confusion around appropriate timing of introducing solids. The absence of a clear FSANZ guideline prohibiting the promotion of foods for infants under 6 months of age undermines current WHO and health professional guidelines. Promotion of CCF products as suitable for infants from 4 months of age, as observed in this study, should be prohibited given the negative effects of early introduction of CCF [37].

Not only does digital marketing by Australian CCF brands normalize and encourage the use of CCF products, it also promotes toddler milks—formula-like products designed for children over 12 months of age—as necessary for the growth and development of children. Numerous studies have shown that consumers do not differentiate between these toddler milks and infant formula, which in Australia is prohibited from advertising under the industry self-regulated MAIF Code [6,38,39,40]. As well as the concern that these products act as proxy advertising for infant formula, the promotion of toddler milks themselves is cause for concern given their expense and associations with overweight and obesity in later life [33].

It has been reported that parents describe infant feeding advice from healthcare practitioners as conflicting or frequently changing [28]. Parents want “factual education related to their individual and personal infant feeding choices, provided with sensitivity, in a non-judgmental manner” [35]. This is what social media marketing of CCF is able to offer parents. Digital marketing is designed to provide tailored content; extensive data collection, combined with sophisticated profiling technology allows for identification of patterns of behaviour that can predict and seemingly anticipate the needs and desires of users [41]. Social media networks provide a platform to facilitate data collection and brand engagement, with incentives like competitions and giveaways, as used by CCF manufacturers here, prompting individuals to create and share promotional content of their own [41]. Large numbers of people are willing to allow corporations into their social networks, providing free advertising for these companies [42]. Research clearly details the potential harms of digital marketing to children and adolescents, noting that even when young users recognise the attempt to influence them, they may still fail to understand and combat its effects [41]. It may be argued that parents constitute a similarly vulnerable population, due to the desire for the best for their child, and the fear of failing to provide this [43].

The use of emotive messaging in advertising is a longstanding marketing tactic. Emotional advertising is designed to increase the likelihood of the desired reaction—often the purchase—and can be used to foster feelings of vulnerability in the target audience [43]. Investigations of food and beverage marketing found that individuals were 2.5 times more likely to prefer brands whose Facebook posts they had a strong positive reaction to [42]. It is clear that Australian CCF brands are playing to this vulnerability, given the content of posts frequently includes aspects designed to be emotionally engaging and featuring desirable attributes of the brand and products. As explained by the theoretical Elaboration Likelihood Model, the combination of logic-driven promotion of positive aspects and imagery of happy, smiling children designed to evoke an emotional response has been seen in previous studies of traditional media sources and is thought to encourage use of the product [31]. Additionally, the effect of emotive messaging to provoke guilt or fear has been identified as a powerful motivator in advertising to parents [43]. These emotional prompts are visible in much of the content of this study. Socially desirable aspects such as self-feeding and making mealtimes a fun experience may be designed to elicit guilt in parents who feel pressure to perform these behaviors. Fear is also used in messaging around products being ‘safe’—particularly with the organic status of foods and promoting fear of additives, pesticides and allergens. Messaging contained in these posts frequently conveys the idea that CCF products are safer, normal and necessary for daily consumption. The relatively unregulated space of digital marketing, particularly social media sites, should be carefully monitored to limit the normalization of reliance on suboptimal practices and CCF products.

This study is not without limitations. The complexity of metrics that control visibility of posts in Facebook news-feeds mean it is not possible to determine what proportion of followers viewed individual posts. The high variability in the number of interactions with similar posts by the same brand indicates that there is wide variation in the number of posts that followers are seeing. Pages are also able to pay for posts to be seen by users who do not follow the page, as well as appearing in feeds of friends of followers, meaning the reach of this marketing is not able to be accurately measured [42]. In addition, results must be considered in light of the events of the coronavirus disease 2019 (Covid-19) pandemic and the subsequent shift in business priorities for many organizations, which may have resulted in the data collection not reflecting the same frequency or content priorities of pre-pandemic times. Furthermore, it is possible that posts or comments may have been removed or altered before or after the point of data collection. Future research in this area is warranted to investigate the impact of these marketing messages and different approaches by brands on parental attitudes. The effect of post format—whether text, image, video or combinations—on message impact may also be an area of interest.

## 5. Conclusions

In Australia, CCF manufacturers utilize social media, predominantly Facebook and Instagram, to promote their brands and products to consumers. Messaging in these communications focuses primarily on positive brand or product attributes such as taste, texture, novelty, self-feeding and solutions to perceived problems; or capitalizing on common parent concerns about organic status of foods, additives and allergens, appropriate texture for age, child development and health. The common strategy of highlighting healthy ingredients and nutrients may create a ‘health halo’ effect and lead parents to mistakenly believe that these products are necessary and preferable to home-prepared foods. The promotion of CCF as suitable for infants under the age of 6 months contravenes WHO recommendations but not FSANZ Standards. It is advisable that these Standards be reviewed to align with the extensive body of evidence that underpins the WHO recommendations. Restriction of both ‘health halo’ messaging and the promotion of toddler milks is also recommended, given the potential impact on parental perceptions of optimal infant feeding. 

## Figures and Tables

**Table 1 ijerph-18-07934-t001:** Commercial complementary food lines by category per outlet.

	Unique Commercial Complementary Food Lines		
	Coles	Woolworths	Chemist Warehouse
	210	147	129
**Product Category**	**Proportion (%)**		
Ready-to-eat Meal	55	69	53
Jar	5	9	0
Larger Pouch	7	12	2
Squeeze Pouch	43	48	51
Ready-to-prepare (Cereal)	4	6	15
Ready-to-cook (Dry Pasta)	0	0	3
Ready-to-eat Snack	41	25	29
First foods (Rusks)	3	5	6
Older Baby/Toddler Snack	38	20	23

**Table 2 ijerph-18-07934-t002:** Brand presence of commercial complementary foods (market share by product lines) in Coles Supermarkets, Woolworths Supermarkets and Chemist Warehouse Pharmacy.

Brand	Product Lines, *n* (%)		
	Coles	Woolworths	Pharmacies
Bellies	20 (9)	23 (16)	-
Baby Mum Mum	4 (2)	3 (2)	5 (4)
Bellamy’s Organic	14 (7)	13 (9)	30 (23)
Bubs Organics	7 (3)	-	22 (17)
CUB	14 (7)	-	-
Farex	2 (1)	4 (3)	5 (4)
Heinz	31 (15)	25 (17)	15 (12)
Kiddylicious	12 (6)	2 (1)	-
Little Quacker	4 (2)	1 (1)	9 (7)
Macro	-	11 (7)	-
Nestle Cerelac	5 (2)	3 (2)	4 (3)
Only Organic	38 (18)	19 (13)	-
Rafferty’s Garden	52 (25)	32 (22)	29 (22)
Whole Kids	7 (3)	2 (1)	10 (8)
Smiling Tums	-	9 (6)	-
Total	210 (100)	147 (100)	129 (100)

**Table 3 ijerph-18-07934-t003:** Facebook posts by commercial complementary food brand pages, 10 June–11 September 2020.

Brand	Page Followers/Page Likes *	Number of Posts		Most Common Post Type (Number Across All Months)
		Average per Month	Total of 3 Months	
Bellies	7198/7022	14.33	43	Product Range (*n* = 19)
Baby Mum Mum	19,649/19,834	2	6	Incentive (*n* = 4)
Bellamy’s Organic	156,377/157,564	4.3	13	Product Range (*n* = 8)
Bubs Organic	85,466/85,199	12	36	Product Range (*n* = 25)
Heinz	48,105/48,780	0	0	-
Kiddylicious	11,748/11,736	8.3	25	Product Range (*n* = 17)
Little Quacker	3083/3088	0.33	1	Product Range (*n* = 1)
Nestle	20,435/20,642	2	6	Support (*n* = 5)
Only Organic	31,783/32,491	6.67	20	Product Range (*n* = 11)
Rafferty’s Garden	58,905/59,316	6.33	19	Product Range (*n* = 9)
Whole Kids	30,350/30,234	16	48	Engagement (*n* = 17)

* Number of page followers and likes as at 12 September 2020.

**Table 4 ijerph-18-07934-t004:** Commercial complementary food brands’ Facebook post categories, marketing message themes and marketing message categories.

	Total Posts All Brands, *n* (%)
**Post Content Category**	
Product Range	104 (48.1)
Broad Values	32 (14.8)
Incentives	23 (10.6)
Social/Parenting Support	11 (5.1)
Engagement/Connection Building	32 (14.8)
Expert/Celebrity Endorsement	10 (4.6)
Informative	4 (1.9)
Total	216
**Marketing Message Themes and Categories**	
**Theme 1—General Product Attributes**	
Taste/Flavor	89 (41.2)
Texture	36 (16.7)
New product	29 (13.4)
**Theme 2—Socially Desirable Attributes**	
Self-Feeding	63 (29.2)
Fun/Positive Experience	41 (18.9)
Solution to a Problem	31 (14.4)
Australian made/owned	11 (5.1)
Happy Children Images	104 (48.1)
**Theme 3—Concern-based Attributes**	
Organic	86 (39.8)
Additive/Allergen Free	39 (18.1)
Referring to Child Age	84 (38.9)
**Theme 4—Health-focused Attributes**	
Child Development/Growth	32 (14.8)
Health Support/Healthy Habit	17 (7.9)
Healthy Ingredients/Nutrition	98 (45.4)

## Data Availability

The data presented in this study are available on request from the corresponding author.

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
