# Peer review of "Digital Marketing of Commercial Complementary Foods in Australia: An Analysis of Brand Messaging"

_ijerph, 2021, doi:10.3390/ijerph18157934_

Round 1
Reviewer 1 Report
Dear Authors,
Thank you for the opportunity to read and review your manuscript submitted to IJERPH. After reading the manuscript I am convinced that the research of digital marketing of commercial complementary foods in Australia is outstanding, relevant, instrumental and valuable. The research itself is very well structured and has a strong methodological basis. However, I offer several recommendations that may improve the quality of the paper:
- The abstract should highlight the purpose of the study, therefore it is recommended to present the purpose in a more clear manner.
- You have collected a valuable database that has a potential for a more detailed insights. My recommendation would be to evaluate differences in post content categories and marketing messages among separate brands / product lines / product categories. Maybe there exist significant differences in brand messaging among brands / product lines / product categories?
- I understand that at the moment the collection of data is already finished, but in the future, it would be interesting to find out how different brand messages reach the potential market (for example, measuring the number of likes, number of shares). Apart from that, it would be profound to compare if there exist differences in brand messages among the separate formats of posts (text, text + picture, picture, video, carousel, etc.).
Once again, thank you for this opportunity to read the manuscript.
Author Response
1. The abstract should highlight the purpose of the study, therefore it is recommended to present the purpose in a more clear manner.
Thank you, the abstract has been altered to present the purpose of the study more clearly (line 10).
2. You have collected a valuable database that has a potential for a more detailed insights. My recommendation would be to evaluate differences in post content categories and marketing messages among separate brands / product lines / product categories. Maybe there exist significant differences in brand messaging among brands / product lines / product categories?
The recommendation to evaluate further the differences in post content and brand messaging is appreciated. It may become the focus for future work.
3. I understand that at the moment the collection of data is already finished, but in the future, it would be interesting to find out how different brand messages reach the potential market (for example, measuring the number of likes, number of shares). Apart from that, it would be profound to compare if there exist differences in brand messages among the separate formats of posts (text, text + picture, picture, video, carousel, etc.).
We agree these would be interesting investigation for future work, particularly the differences in messaging with format. This has also been noted as a direction for future research in the paper (line 389).
Thank you for your review of our work. We appreciate your comments.
Reviewer 2 Report
The title is confusing because it includes "mixed methods" but the article does not deal with quantitative methods as such. Quantitative methods involve testing hypotheses using statistical techniques, something that is not done in this work.
The part of quantitative analysis (Websites and Social Media) that appears in 2. Materials and Methods is only a description of how the brands were identified and the information was captured. This mere description does not imply the use of any quantitative methodology.
Qualitative analysis could be improved if relationships between variables are studied. For example, the post content categories (Table 4) could be crossed with the information of the categories of products (Table 1), brands and supermarkets (Table 2). Perhaps this would enrich the results presented.
Author Response
1. The title is confusing because it includes "mixed methods" but the article does not deal with quantitative methods as such. Quantitative methods involve testing hypotheses using statistical techniques, something that is not done in this work. The part of quantitative analysis (Websites and Social Media) that appears in 2. Materials and Methods is only a description of how the brands were identified and the information was captured. This mere description does not imply the use of any quantitative methodology.
Thank you, this has been amended.
2. Qualitative analysis could be improved if relationships between variables are studied. For example, the post content categories (Table 4) could be crossed with the information of the categories of products (Table 1), brands and supermarkets (Table 2). Perhaps this would enrich the results presented.
As the term mixed methods has now been removed, it does not seem suitable to assess the relationships between these variables. However this type of analysis may be useful for future work, thank you.
Reviewer 3 Report
The article ‘Digital Marketing of Commercial Complementary Foods in Australia: a Mixed Methods Analysis of Brand Messaging’, presented for review is very interesting. However, some issues need to be clarified or supplemented. The comments are included below. Some of the comments are debatable.
- What are the recommendations for breastfeeding in Australia? Complementary nutrition should be adapted to the child. For some fast-growing babies, it may not be sufficient to breastfeed for 6 months.
- Initial nutrition preparations were not investigated in the study, but whether such products contain information that breastfeeding gives additional benefits.
- Do manufacturers inform that the introduction of supplementary products should be consulted by parents with a pediatrician?
- Did the research take into account that food for young children is food for children under three years of age? There are special requirements for such foods. Does Australian food law take this into account?
- Do parents know why factory-made complementary products are better and safer than home-made ones? Do producers inform parents in this direction?
- What is the parent's nutritional awareness? Do parents know that baby food undergoes rigorous testing and is produced from selected raw materials.
- Has it been taken into account that negative posts can be removed by the site administrator? Has this issue been included in the research?
Author Response
The article ‘Digital Marketing of Commercial Complementary Foods in Australia: a Mixed Methods Analysis of Brand Messaging’, presented for review is very interesting. However, some issues need to be clarified or supplemented. The comments are included below. Some of the comments are debatable.
- What are the recommendations for breastfeeding in Australia? Complementary nutrition should be adapted to the child. For some fast-growing babies, it may not be sufficient to breastfeed for 6 months.
The Australian Infant Feeding Guidelines (IFGs) are similar to the WHO recommendations in that they recommend “…exclusive breastfeeding to around 6 months of age” and then “…continue breastfeeding while introducing appropriate solid foods…” We agree that complementary nutrition should be adapted to the child, and the IFGs allow for this with the phrase “around 6 months” and advising that that parents look for signs of readiness. However, in Australia early introduction of solid foods is an issue, with approx. 35% of babies receiving solid food at 4 months of age and 92% by 6 months. We have clarified this in the paper at line 50.
- Initial nutrition preparations were not investigated in the study, but whether such products contain information that breastfeeding gives additional benefits.
The limited research in this area to date has been on breastfeeding and breastmilk substitutes (see Harris & Pomeranz 2020), so they were not the focus of this study. However, in the Australian Food Standards Code it is a requirement that all Infant formula products display the statement, “Breast milk is best for babies. Before you decide to use this product, consult your doctor or health worker for advice.” (Standard 2.9.1-19.1b).
- Do manufacturers inform that the introduction of supplementary products should be consulted by parents with a pediatrician?
Within the Australian Food Standards Code manufacturers of infant foods “must not include a recommendation, whether express or implied, that the food is suitable for infants under the age of 4 months” (Standard 2.9.2-7.2), they must also specify a minimum age (Standard 2.9.2-7.3b) and if the age is less than 6 months they must also include the statement “‘Not recommended for infants under the age of 4 months” (Standard 2.9.2-7.3b). But there is no provision about stating that parents should consult a medical professional before introducing solids.
- Did the research take into account that food for young children is food for children under three years of age? There are special requirements for such foods. Does Australian food law take this into account?
In Australia, there are no special food laws for children over 12 months of age, which is a real gap in our food regulation (see discussion section 4). That is why we recommend extending the current standard that prohibits health and nutrition content claims for infant formula products to cover these commercial complementary foods as well as formula for children over 12 months (section 4). As you note, to three years would be ideal. In our research, we used the “baby aisle” at the supermarket as our natural indicator of what is considered food for young children (methods, section 2). This supermarket aisle includes nappies, formula, bottles, spoons, bibs and other feeding equipment and is set up as an aisle that parents of young children would walk down at almost every shop, and that no one else would ever need to visit. Our approach has been clarified in methods (line 117).
- Do parents know why factory-made complementary products are better and safer than home-made ones? Do producers inform parents in this direction?
There is little research to date in this area, however it appears parents are frequently misleadingly informed that factory-made products are superior to home-made foods, despite this contradicting WHO recommendations and the evidence of other research (see Dunford et al 2015, ) regarding the suboptimal nutritional and textural properties of CCF products.
- What is the parent's nutritional awareness? Do parents know that baby food undergoes rigorous testing and is produced from selected raw materials.
As we were only looking at the food products and digital marketing strategies, the understanding of parents is beyond the scope of this study. Previous research indicates parents receive much conflicting information from a multitude of sources regarding feeding their children (see Adamo & Brett 2014, Begley et al 2019). They seem to have quite high levels of trust in manufacturer products, especially those that use words like “organic” in the marketing (Begley et al 2019). Overall, parent concern about the quality of the food supply is high, but this affects ingredients in both commercial and home-made complementary foods (Begley et al 2019).
- Has it been taken into account that negative posts can be removed by the site administrator? Has this issue been included in the research?
This research only included posts that were under administrator control (see materials and methods line 145). We have no way to know what, if any comments have been removed, nor whether any posts have been deleted. This has been added as a limitation at line 386.